# Differences in Physiological Characteristics, Seed Germination, and Seedling Establishment in Response to Salt Stress between Dimorphic Seeds in the Halophyte *Suaeda liaotungensis*

**DOI:** 10.3390/plants12061408

**Published:** 2023-03-22

**Authors:** Jieqiong Song, Hongfei Wang, Ruowen Chu, Lantong Zhao, Xinxin Li, Shuo An, Mengke Qiang, Wanying Du, Qiuli Li

**Affiliations:** Key Laboratory of Plant Biotechnology of Liaoning Province, School of Life Sciences, Liaoning Normal University, Dalian 116081, China

**Keywords:** *Suaeda liaotungensis*, dimorphic seeds, salt stress, physiological index, seed germination, seedling growth

## Abstract

Soil salinization is an increasing agricultural problem around the world, affecting crop productivity and quality. Seed germination and seedling establishment are susceptible to salt stress. *Suaeda liaotungensis* is a halophyte with strong salt tolerance that produces dimorphic seeds to adapt to the saline environment. Differences in physiological characteristics, seed germination, and seedling establishment in response to salt stress between dimorphic seeds in *S. liaotungensis* have not been reported. The results showed that brown seeds had significantly higher H_2_O_2_ and O_2_^−^. levels and betaine content, as well as POD and CAT activities, while they had significantly lower MDA and proline contents and SOD activity than black seeds. Light promoted the germination of brown seeds in a certain temperature range, and brown seeds could reach a higher germination percentage in a wide temperature range. However, light and temperature had no effect on the germination percentage of black seeds. Brown seeds had higher germination than black seeds under the same NaCl concentration. The final germination of brown seeds was significantly decreased as salt concentration increased, whereas this had no effect on the final germination of black seeds. POD and CAT activities, as well as MDA content, in brown seeds were significantly higher than those in black seeds during germination under salt stress. Additionally, the seedlings from brown seeds were more tolerant to salinity than those from black seeds. Therefore, these results will give an in-depth understanding of the adaptation strategies of dimorphic seeds to a salinization environment, and better exploitation and utilization of *S. liaotungensis*.

## 1. Introduction

Soil salinization is an increasing agricultural problem around the world, affecting crop yield and quality. Plants can suffer from ion imbalance and osmotic stress when exposed to high salinity, which results in the growth of most plants being inhibited to varying degrees and even unable to grow and survive [1]. Excessive ions can cause a reduction of cell expansion, damage cell structure, and inhibit the activities of various enzymes in plant cells [2]. Osmotic stress can prevent plant roots from taking water from the outside, which causes dehydration and the death of plant cells. A saline environment also causes the disorder of oxygen metabolism in plants, which increases the formation of reactive oxygen species (ROS) and inhibits the activity of many essential enzymes. Excessive ROS induce the peroxidation of membrane lipids and destroy the nucleic acids, proteins, and other biological macromolecules in plants, thus inhibiting the growth of plants [2,3].

Seed heteromorphism of halophytes is an important feature of adaptation to adverse environments such as drought, desert, and salinization [4]. Seed heteromorphism has been reported in many halophytes, including *Chenopodium*, *Salicornia*, *Salsola*, *Suaeda*, and *Trianthema* [5,6,7]. There are significant differences in morphological structure, dormancy and germination characteristics, and salt tolerance among heteromorphic seeds, which provides a variety of alternative opportunities for plant population formation [8,9,10].

Seed germination and seedling growth are the key stages in the life history of plants, which are susceptible to environmental factors. Light is a significant environmental factor, and the impact of light on seed germination varies on the type of plant and other germination circumstances [11]. Light has little effect on the germination of non-dormant brown seeds of *Suaeda aralocaspica*, but it can stimulate the germination of cold-stratified black seeds [12]. For the time of seed germination and seedling establishment in desert plants, temperature is essential [13]. Compared to dormant seeds, non-dormant seeds have a wider temperature range for germination [13,14]. High salinity can inhibit seed germination and delay the time of germination [15,16]. In addition, the ionic composition in soil also affects seed germination and seedling development [14,17]. At present, there have been many studies on heteromorphic seed germination and seedling establishment, but most have mainly focused on environmental factors such as light, temperature, and salt [18,19,20].

*Suaeda liaotungensis* (Kitag), an annual herbaceous plant belonging to the Amaranthaceae family, is found in coastal saline–alkali soils in Dalian, China. *Suaeda* plants possess the ability to absorb heavy metals to improve heavy metals-contaminated soil, thereby reflecting its ecological value [21]. According to the investigation, it was found that *S. liaotungensis* produced brown and black seeds and exhibited seed dimorphism. Current research focuses mostly on the expression of salt-tolerant genes and rhizosphere soil microorganism in *S. liaotungensis* [22,23,24,25]. The differences in germination and physiological changes between dimorphic seeds and their relationship with salt tolerance in *S. liaotungensis* remain unclear.

This study aimed to test the differences in physiological indexes between dimorphic seeds, as well as test the effects of light, temperature, and salinity on the germination of dimorphic seeds and the effects of salinity on seedling and plant growth in *S. liaotungensis*. In this study, we asked the following questions: (1) What is the relationship between physiological indexes’ difference and salt tolerance of dimorphic seeds? (2) Do dimorphic seeds require different conditions for light and temperature during germination? (3) Which type of seed has a higher salt tolerance during the germination stage? (4) Is there a difference in the range of salt tolerance between the two types of seeds? The findings will give an in-depth understanding of the adaptation strategies of dimorphic seeds to a salinization environment, and better exploitation and utilization of *S. liaotungensis.*

## 2. Results

### 2.1. Morphological Characteristics of Dimorphic Seeds in S. liaotungensis

The seed heteromorphism including brown and black seeds was found in *S. liaotungensis* (Figure 1A). The brown seed was enclosed by a thin membranous seed coat and has non-dormant characteristics (Figure 1A). The black seed was enclosed by a thick leathery seed coat and has non-deep physiological dormancy (Figure 1A). The length, width, thickness, and mass of dimorphic seeds were measured, and it was found that there were significant differences in thickness and mass of 1000 seeds between dimorphic seeds (Figure 1B).

### 2.2. Difference in ROS Levels from Dimorphic Seeds in S. liaotungensis

Previous research has shown that ROS is a key indicator affecting seed germination and stress tolerance in plants [26,27,28]. ROS may be related to seed vigor, and it can be used as a signaling molecule to break seed dormancy. The results showed that the H_2_O_2_ and O_2_^−^. levels in brown seeds were significantly higher than those in black seeds (Figure 2A,B). Malondialdehyde (MDA) can be used as an indicator of membrane lipid peroxidation [27]. The MDA content of black seeds was found to be 1.62 times higher than that of brown seeds (Figure 2C).

### 2.3. Difference in Antioxidant Enzyme Activity from Dimorphic Seeds in S. liaotungensis

Based on changes in ROS levels, we further detected the activities of antioxidant enzymes such as superoxide dismutase (SOD), peroxidase (POD), and catalase (CAT) in dimorphic seeds. SOD specifically catalyzes O_2_^−^. to form H_2_O_2_ and O_2_, and then POD and CAT catalyze H_2_O_2_ to form H_2_O [29]. As shown in Figure 3A, black seeds exhibited significantly higher SOD activity than brown seeds. However, the POD and CAT activities of brown seeds were 1.55 and 1.43 times higher, respectively, than those of black seeds (Figure 3B,C).

### 2.4. Difference in Osmotic Regulating Substances from Dimorphic Seeds in S. liaotungensis

Osmotic adjustment substances, such as betaine and proline, are involved in seed germination and salt tolerance. Under salt stress, the contents of these substances accumulate rapidly to maintain the water potential of cells. Therefore, we detected the betaine and proline contents of dimorphic seeds. The betaine content of brown seeds was 2.6 times that of black seeds (Figure 4A), while the proline content was significantly lower (Figure 4B).

### 2.5. Effects of Light and Temperature on Seed Germination in S. liaotungensis

Light, temperature, moisture, and seed type all affect seed germination. Under the same conditions, brown seeds had a significantly higher germination percentage than black seeds (Figure 5). From 15 °C to 25 °C, the germination percentages of brown seeds in light were significantly higher than in darkness (Figure 5A), suggesting that light could promote the germination of brown seeds. For black seeds, there was no discernible difference in germination under light and darkness conditions, suggesting that light had no effect on the germination of black seeds (Figure 5B). Brown seeds had a higher germination percentage, germination index, and germination potential than black seeds at the same temperature when exposed to light (Figure 5C–E). There was no discernible difference in the germination of dimorphic seeds with varied temperature treatments (Figure 5C). As temperature increased, the germination indexes of dimorphic seeds increased (Figure 5D). Except for 15 °C, the germination potential of dimorphic seeds did not substantially alter with different temperature treatments (Figure 5E).

### 2.6. Effect of Salinity on Seed Germination in S. liaotungensis

The germination percentages of dimorphic seeds decreased as NaCl concentration increased (Table 1). Brown seeds had significantly higher germination and final germination percentages than black seeds under the same NaCl concentration treatment. Brown seeds germinated at 61.3% in an 800 mM NaCl solution, while the percentage of black seeds was only 4%. At a 1400 mM NaCl condition, the brown seeds still germinated up to 19.3%, while the black seeds did not germinate at all. The final germination percentage of brown seeds reached a high level, whereas that of black seeds was only about half as high. The germination recovery percentage of dimorphic seeds exhibited an increasing trend as NaCl concentration increased, but there was no discernible difference between dimorphic seeds except in a 400 mM NaCl solution.

The germination indexes of dimorphic seeds declined as NaCl concentration increased (Table 2). At the same NaCl condition, brown seeds had a higher germination index than black seeds (Table 2). For brown seeds, the maximum germination index was 43.3, while the lowest germination index was 5.3 in NaCl solutions. For black seeds, the maximum germination index was 18.4, while the lowest germination index was 0 in NaCl solutions (Table 2).

### 2.7. Effect of Salinity on the Physiological Indexes of Dimorphic Seeds during Germination in S. liaotungensis

Brown seeds had significantly higher POD activity than black seeds, but there was no significant difference in CAT activity between dimorphic seeds during germination in the absence of NaCl solution (Figure 6A,B). In the presence of NaCl solution, brown seeds had significantly higher POD and CAT activities than black seeds during germination (Figure 6A,B). Compared to the control, POD and CAT activities in brown seeds increased, whereas those in black seeds decreased during germination under salt stress (Figure 6A,B). The MDA and proline contents in brown seeds were significantly lower than those in black seeds during germination in the absence of NaCl solution (Figure 6C,D). In the presence of NaCl solution, brown seeds had significantly higher MDA content and lower proline content than black seeds during germination (Figure 6C,D). Compared to the control, the MDA and proline contents in brown seeds increased during germination under salt stress (Figure 6C,D). Compared to the control, the MDA content in black seeds decreased, whereas proline content in black seeds increased during germination under salt stress (Figure 6C,D).

### 2.8. Effect of Salinity on Seedling Growth in S. liaotungensis

Salt stress had an impact on both seed germination and seedling growth. Seedling growth of dimorphic seeds was significantly inhibited as NaCl concentration increased (Figure 7A). At the same NaCl concentration, the radicle and shoot lengths of seedlings from brown seeds were higher than those from black seeds (Figure 7B–E). When compared to distilled water, the shoot lengths of seedlings from brown seeds significantly increased with 100–200 mM NaCl solutions (Figure 7B,C). After the 800 mM NaCl treatment, the radicle and shoot growth of dimorphic seeds were significantly inhibited. The radicle and shoot lengths of brown seeds were 5.3 and 7.6 mm, respectively, while the radicle and shoot growth of black seeds were entirely prevented at 1000 mM NaCl solution (Figure 7B–E).

The radicle tolerance indexes of dimorphic seeds generally showed declining trends as NaCl concentration increased (Table 3). The shoot tolerance index of brown seeds with 100–200 mM NaCl solution showed a significantly increase, whereas that of black seeds had no significant difference (Table 4). After the 800 mM NaCl treatment, browns seeds had significantly higher radicle and shoot tolerance indexes compared to black seeds at the same condition (Table 3 and Table 4). Additionally, the shoot tolerance index was higher than the radicle tolerance index of the same seed type at the same NaCl concentration (Table 3 and Table 4).

### 2.9. Effect of Salinity on Growth of Plants in S. liaotungensis

The growth of plants from dimorphic seeds was significantly promoted as NaCl concentration increased (Figure 8 and Figure 9). For the plants from brown seeds, compared to the control, plant height and leaves and nodes numbers of plants with 100–200 mM NaCl solution significantly increased, but plant height at 400 mM NaCl solution showed a declining trend, while the leaves and nodes numbers significantly decreased at 600 mM NaCl solution (Figure 8A–C). The fresh and dry weights of plants (shoot, root, and total) significantly increased with 100–400 mM NaCl solution, while those of root also significantly increased with 600 mM NaCl solution compared to the control (Figure 8D–I).

For the plants from black seeds, plant height with 100–200 mM NaCl solution significantly increased but showed a slight decline at 400 mM NaCl solution (Figure 9A). Compared to the control, the leaves and nodes numbers of plants with 100–400 mM NaCl solution significantly increased but there was no significant difference at 600 mM NaCl solution (Figure 9B,C). The fresh and dry weights of plants (shoot, root, and total) significantly increased with 100–600 mM NaCl solution compared to the control (Figure 9D–I).

Under 200 mM NaCl treatment, there were no discernible variations in plant height (*p* = 0.409), number of leaves (*p* = 0.336), number of nodes (*p* = 0.609), fresh weight of shoot (*p* = 0.557), fresh weight of root (*p* = 0.741), total fresh weight (*p* = 0.535), dry weight of shoot (*p* = 0.945), dry weight of root (*p* = 0.179), and total dry weight (*p* = 0.936) of the plants from dimorphic seeds (Figure 7 and Figure 8). However, the fresh and dry weights of plants (shoot, root, and total) from brown seeds were larger than those from black seeds. Additionally, the growth of plants under 200 mM NaCl treatment was promoted most obviously, suggesting that the optimal salt concentration for growth of plants from dimorphic seeds was 200 mM.

## 3. Discussion

Seed heteromorphism is the result of halophyte long-term adaptation to the saline environment. Dimorphic seeds are common in Amaranthaceae. Large seeds have light color and no dormancy, while small seeds have dark color and deep dormancy [30,31]. In this study, *S. liaotungensis* also had similar seed heteromorphic characteristics. Dimorphic seeds differed greatly in size and the color and texture of the seed coat (Figure 1), which may be responsible for the different dormancy and germination characteristics of dimorphic seeds.

Many studies have shown that physiological active substances in seeds have a great influence on seed germination and salt tolerance [32,33,34,35]. Excessive ROS is not conducive to plant growth, but ROS levels are kept at a level that causes cellular activities related to germination, such as hormone signaling [36]. It has been reported that the hydroxyl radical (.OH) mediates cell wall loosening during germination, thereby promoting seed germination [37]. In this study, brown seeds had larger levels of O_2_^−^. and H_2_O_2_ than black seeds (Figure 2A,B), and their germination rate was significantly higher than that of black seeds (Figure 5C). These results suggest that the presence of higher basal levels of ROS should be an indication of greater germination, regardless of the environmental condition. Moreover, brown seeds had a significantly lower MDA level than black seeds in *S. liaotungensis* (Figure 2C), suggesting that MDA can be used as an indicator of salt tolerance. Black seeds had significantly higher SOD activity but lower POD and CAT activities than brown seeds in *S. liaotungensis* (Figure 3), which is similar to the findings for heteromorphic seeds of *Chenopodium album*, but differs from the reported result that under non-salt stress, black seeds have higher CAT activity than brown seeds [38]. SOD, POD, and CAT play similar or different functions to resist salt stress [34]. Therefore, a higher antioxidant enzyme activity in brown seeds provided favorable conditions for germination under salt stress, which may also be one of the reasons why brown seeds have a higher ability for germination and salt resistance than black seeds. The osmotic regulatory substance betaine can alleviate the oxidative damage caused by salt stress through protecting the activities of antioxidant enzymes in plants [39]. Brown seeds had significantly higher betaine content than black seeds in *S. liaotungensis* (Figure 4A), suggesting that betaine might act as a positive regulator in the salt-tolerance of dimorphic seeds in *S. liaotungensis*. Proline is also another important osmoprotectant, and its accumulation can reduce stress damage to plant cells, thus enhancing the stress tolerance of plants [40,41]. In this study, black seeds had a higher proline content than brown seeds (Figure 4B), suggesting that proline might play roles in salt tolerance with various osmotic regulatory substances.

Seed germination is susceptible to external environmental factors such as light, temperature, and salinity. Light promoted the germination of brown seeds in certain temperature ranges, but there was no effect on the germination percentage of black seeds (Figure 5A,B). Brown seeds could reach a higher germination percentage in a wide temperature range; however, the germination temperature range of black seeds was relatively narrow, and black seeds had a higher germination rate at 25 °C (Figure 5A,B), which suggests that they have non-deep dormancy characteristics. At the same temperature, brown seeds had a higher germination percentage, germination index, and germination potential than black seeds (Figure 5C–E). The reason for the low germination rate of black seeds may be their thicker seed coat which limited radicle elongation to a certain extent, or the low content of germination-promoting hormones in black seeds, which needs further investigation. Therefore, the difference in germination time of dimorphic seeds reduced the risk of one-time destruction, thereby improving the continuity of the population in *S. liaotungensis*. The germination percentages of dimorphic seeds decreased as salinity increased. Brown seeds germinated significantly higher than black seeds at the same salt concentration, which was similar to that of *Suaeda salsa* [14,42]. Under 1400 mM NaCl treatment, the germination of brown seeds was still 19.3%, while the black seeds did not germinate at all (Table 1), suggesting that brown seeds were more salt-tolerant than black seeds at the germination stage. Our results showed that POD and CAT activities, as well as MDA content, in brown seeds were significantly higher than those in black seeds during germination in the presence of salinity (Figure 6A–C), which indicated that higher salt tolerance of brown seeds at the germination stage was related to these physiological active substances. Moreover, compared to the control, POD and CAT activities, as well as MDA and proline contents, in brown seeds significantly increased during germination under salt stress (Figure 6), which was one of the reasons why brown seeds still have a higher germination rate under salt stress. POD and CAT activities, as well as MDA content, in black seeds significantly decreased during germination under salt stress (Figure 6A–C), which would lead to a lower germination rate of black seeds. After transferring to deionized water, a germination recovery experiment showed that the percentage of black seeds that germinated steadily increased, and the recovery germination rate of brown seeds was lower than that of black seeds (Table 1), which is consistent with results reported in other halophytes [10,14,43]. Moreover, brown seeds had a significantly higher germination index than black seeds at the same salt concentration (Table 2). These differences allowed the two types of seeds to select their own favorable salt environment and the time for germination, thereby improving progeny survival [44].

Seed heteromorphism extends the time for germination, which is beneficial to the establishment of seedlings. It has been shown that the seedlings from heteromorphic seeds of halophytes show significant differences in salt tolerance, and seedling growth from black seeds was significantly inhibited, but those from brown seeds was not affected under 400 mM NaCl treatment in *Suaeda salsa* [14]. Our findings also demonstrated that the seedlings from brown seeds are more salt-tolerant than those from black seeds in *S. liaotungensis* (Figure 7). Studies have found that antioxidants’ enzyme activities in salt-tolerant varieties are significantly higher than those in salt-sensitive varieties [45,46]. Exogenous application of betaine increases the tolerance of rice seedlings to salt-induced oxidative damage [47,48]. Changes in the membrane lipid composition of seedlings may enhance the salt tolerance [49]. The salt tolerance of seedlings was closely related to the endogenous physiological active substances, but the difference in physiological active substances of seedlings from dimorphic seeds in *S. liaotungensis* needs further investigation. Additionally, the seedlings of brown seeds had considerably greater radicle and shoot tolerance indexes than black seeds (Table 3 and Table 4). In early spring, the low rainfall leads to a high salinity in soil. Most brown seeds begin to germinate, and seedlings from brown seeds emerge under high salinity. Seedling growth from brown seeds is susceptible to high salinity, which is a high-risk strategy. However, when the temperature rises above 0 °C in early spring, the black seeds begin the stratification process by absorbing the melted snow and ice in soil. The dormancy is slowly released and the black seeds begin to germinate. Black seeds had lower tolerance indexes of radicle and shoot, which can protect seedlings from black seeds against salt stress. As the temperature and precipitation gradually increased, the soil salinity decreased. The seedlings from black seeds emerge under low salinity, which is a low-risk strategy. This bet-hedging strategy is beneficial for seedlings of dimorphic seeds to adapt to the saline environment.

Our results also showed that plant growth from dimorphic seeds was stimulated by the increase of NaCl concentration, but there was no significant difference between the plants from dimorphic seeds with tested physiological parameters (Figure 8 and Figure 9), which is similar to the results shown from the dimorphic seeds in *Suaeda splendens* [50]. Therefore, seed dimorphism only influenced seed germination and seedling establishment, but there was no effect on the growth of plants from dimorphic seeds. The 200 mM NaCl condition showed an optimal growth of plants from dimorphic seeds (Figure 8 and Figure 9), which is similar to the results reported in *Suaeda salsa* [51]. Therefore, appropriate salinity is required during seed maturation for seedling emergence and population establishment in halophytes.

## 4. Materials and Methods

### 4.1. Plant Materials

Seeds of *S. liaotungensis* were collected at 121.36°/38.99° (longitude/latitude) from coastal saline–alkali soils in Yingchengzi Town, Dalian City, Liaoning Province, China, in November 2020. The habitat of this species is a temperate sub-humid continental monsoon climate with marine climate characteristics. The mean annual temperature is 10 °C, the highest summer temperature 35 °C, and the lowest winter temperature −18 °C. Annual precipitation is 550–1000 mm. *S. liaotungensis* is the single population with no other species in this habitat. Seeds germinate in March and mature in October. Collected seeds were dried in a laboratory and stored at 4 °C in a refrigerator for subsequent experiments.

### 4.2. Morphological Characteristics of Dimorphic Seeds

Microscopic images of dimorphic seeds from *S. liaotungensis* were obtained with a stereo fluorescence microscope (Leica M165 FC, Wetzlar, Germany). The length, width, and thickness of 50 brown seeds and 50 black seeds were measured with a caliper. The 1000-seed mass of dimorphic seeds was weighed using a balance.

### 4.3. Determination of Physiological Indexes

The determination of hydrogen peroxide (H_2_O_2_) content was slightly modified according to the method of Brennan and Frenkel (1977) [52]. A 0.2 g quantity of seeds was ground in an ice bath of 3 mL precooled acetone. The homogenate was centrifuged and the supernatant was mixed with 20% (*v*/*v*) titanium chloride (TiCl_4_) and concentrated ammonia water. The precipitate was dissolved with 2 M H_2_SO_4_ after centrifugation. The absorbance value at 415 nm wavelength was determined with a spectrophotometer (Genova, British).

The determination of superoxide anion (O_2_^−^.) content was slightly modified according to the method of Panda (2007) [53]. A 0.2 g quantity of seeds was homogenized in pre-cooled sodium phosphate buffer (50 mM, pH 7.8) and centrifuged at 12,000 rpm for 20 min. The supernatant was mixed with 0.8 mL of sodium phosphate buffer and 0.2 mL of hydroxylamine hydrochloride (10 mM), and then placed in a warm bath at 25 °C for 1 h. A 1 mL volume of 17 mM ρ-aminobenesulfonic acid and 7 mM α-naphthylamine were added and shaken evenly, and warmed in the bath at 25 °C for 20 min. The absorbance value at 530 nm wavelength was determined with a spectrophotometer (Genova, British).

A quantity of 0.2 g seeds for each sample was homogenized in a phosphate buffer solution (PBS) and centrifuged at 4 °C for 30 min (12,000 rpm/min for SOD, 6000 rpm/min for POD and CAT), and the supernatant was designated as the crude extract of antioxidant enzymes. The activities of superoxide dismutase (SOD), peroxidase (POD), and catalase (CAT) were detected according to the reported method [54].

The MDA content was detected according to the reported method [55]. A 0.2 g quantity of seeds was homogenized in 5 mL of 10% (*w*/*v*) trichloroacetic acid (TCA). After centrifugation, the supernatant was shaken with 2 mL of 0.6% (*w*/*v*) 2-thiobarbituric acid and then placed in boiling water (95 °C) for 15 min before being rapidly placed in ice to terminate the reaction. The spectrophotometer was used to detect absorbance at 532 nm, 600 nm, and 450 nm wavelengths.

For the determination of betaine content, 0.2 g seeds were homogenized in 10 mL betaine extract (methanol: chloroform: distilled water = 12: 5: 3). The homogenate was shaken in a water bath at 65 °C for 10 min. After centrifugation, the supernatant was dried at 70 °C and dissolved in 3 mL of distilled water to obtain a crude enzyme extract. Betaine content was measured using a betaine assay kit (Solarbio, China) and the absorbance value at 525 nm wavelength was determined with a spectrophotometer (Genova, British).

Proline content was detected according to a previously reported approach [56].

### 4.4. Seed Germination

Quantities of 50 brown seeds and 50 black seeds were chosen, respectively, and placed in Petri dishes covered with 2 layers of filter paper soaked in 10 mL of distilled water or different concentrations of NaCl. The Petri dishes were transferred to a plant incubator (GZX-300BS). To test the effect of light on germination, the light condition was set to 12 h light/12 h darkness or constant darkness. Temperature regimes of 15 °C, 20 °C, 25 °C, 30 °C, and 35 °C were used to test the effect of temperature on germination. When radical protrusion was visible, seeds were deemed germinated. Germinated seeds were scored every day. Seeds were incubated for 10 days under a light condition (1500 Lux) or constant darkness. Three biological replicates were carried out. For each replicate, 3 Petri dishes were used with 50 seeds each.

To test the effect of salinity on germination and recovery, 50 brown seeds and 50 black seeds, respectively, were placed in the Petri dish according to the method described above. Sterile water and different concentrations of NaCl solution (100, 200, 400, 800, 1000, 1400 mM) were added, respectively. The light condition was set to 12 h light/12 h darkness and temperature was set to 20 °C. After 10 days, ungerminated seeds were rinsed before being transferred to Petri dishes with distilled water under the above culture conditions for 7 days. The germinated seeds were scored every day. Three biological replicates were carried out. For each replicate, 3 Petri dishes were used with 50 seeds each. The germination percentage, recovery germination percentage, and final germination were calculated according to the reported method [14].

### 4.5. Seedling Growth

According to the germination time of dimorphic seeds under different NaCl concentrations, 30 germinated seeds were chosen and transferred to the new Petri dishes with different concentrations of NaCl solution (100, 200, 400, 800, 1000, 1400 mM) and cultured for 10 days. The seedlings from the dimorphic seeds were photographed. Three biological replicates were carried out. For each replicate, the radicle and shoot lengths of 10 seedlings were measured. The radicle and shoot tolerance indexes were calculated according to the reported method [14].

### 4.6. Plant Growth

The sand culture method was used to detect the effect of salinity on the growth of plants [57]. The NaCl concentration gradients were set as 100, 200, 400, and 600 mM. In total, 50 brown seeds and 50 black seeds germinated in different NaCl solutions for 7 days. These seedlings of the same size were transplanted into the nutrient bowl which was filled with the same weight of sandy soil. The same amount of NaCl solution was added to the sand until the soil was completely wet without solution outflow. After weighing the initial weight of each bowl, the seedlings were incubated in the culture chamber at 20 °C with light treatment for 30 days. Every 3 days, the distilled water was added to ensure that its weight was the same as the initial weight. After 30 days, these plants were removed from the nutrient bowl and the sand soil adhered to the roots was washed. Then, 15 plants were randomly selected to measure the relevant morphological indexes.

### 4.7. Statistical Analysis

All data were presented as means ± SD from three biological replicates. SPSS 25.0 software was used for statistical analysis. One-way ANOVA followed by an LSD test were used to identify significant differences in the effects of temperature on seed germination and salinity on germination, seedlings, and plant growth. An independent samples *t*-test was used to compare the physiological indexes and germination between dimorphic seeds in the same conditions.

## 5. Conclusions

Seed heteromorphism is the result of halophyte long-term adaptation to the saline environment. There were differences in physiological characteristics, seed germination, and seedling establishment in response to salt stress between dimorphic seeds in *S. liaotungensis*. Brown seeds had higher ROS levels, POD and CAT activities, and betaine content, as well as lower SOD activity and MDA and proline contents, than black seeds. There were significant differences in response to light, temperature, and salinity in dimorphic seeds. Light and temperature promoted the germination of brown seeds, whereas they had no effect on the germination of black seeds. Brown seeds had significantly higher POD and CAT activities, as well as MDA content, than black seeds during germination under salt stress, which allowed brown seeds to have a higher germination rate under salt stress. The seedlings from brown seeds had longer radicle and shoot lengths, as well as higher radicle and shoot tolerance indexes, than black seeds under high salinity. However, salt stress had no obvious effect on plant growth from dimorphic seeds. The findings lay the foundation for understanding the ecological adaptation mechanism of the seeds in *S. liaotungensis*.

## Figures and Tables

**Figure 1 plants-12-01408-f001:**
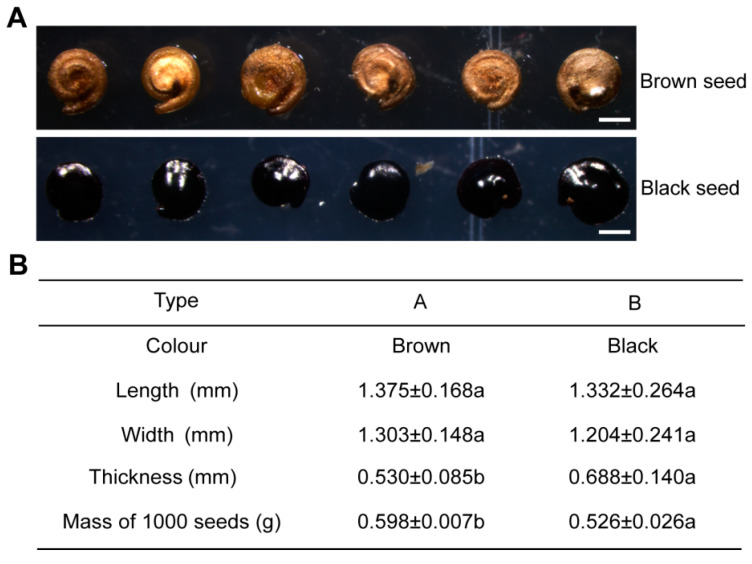
Morphological characteristics of dimorphic seeds of *S. liaotungensis*. (**A**) Microscopic images of the dimorphic seeds. Bars = 1 mm. (**B**) Comparison of length, width, thickness, and 1000-seed mass of dimorphic seeds. There were 50 seeds counted for each replicate. Different letters show significant differences (*t*-test, *p* < 0.05).

**Figure 2 plants-12-01408-f002:**
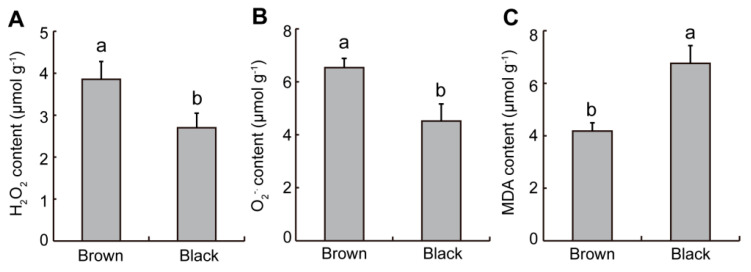
Determination of ROS levels in dimorphic seeds of *S. liaotungensis*. (**A**) H_2_O_2_ content. (**B**) O_2_^−^. content. (**C**) MDA content. Different letters show significant differences (*t*-test, *p* < 0.05).

**Figure 3 plants-12-01408-f003:**
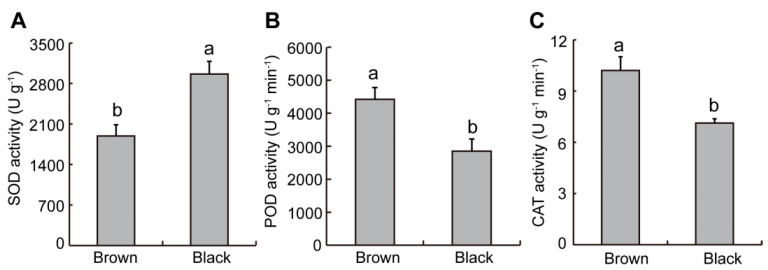
Determination of antioxidant enzyme activity in dimorphic seeds of *S. liaotungensis*. (**A**) SOD activity. (**B**) POD activity. (**C**) CAT activity. Different letters show significant differences (*t*-test, *p* < 0.05).

**Figure 4 plants-12-01408-f004:**
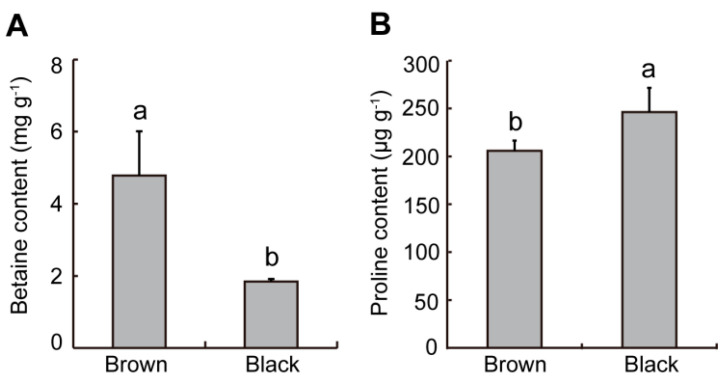
Determination of osmotic regulating substances in dimorphic seeds of *S. liaotungensis*. (**A**) Betaine content. (**B**) Proline content. Different letters show significant differences (*t*-test, *p* < 0.05).

**Figure 5 plants-12-01408-f005:**
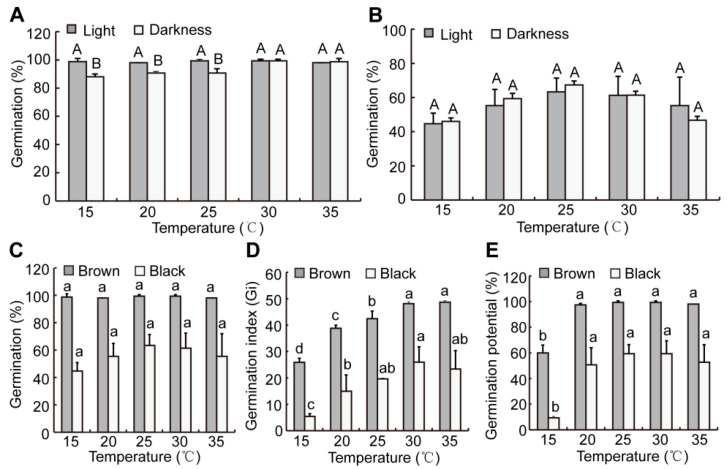
Effects of light and temperature on germination of brown (**A**) and black (**B**) seeds in *S. liaotungensis*. Effect of temperature on germination (**C**), germination index (**D**), and germination potential (**E**) of dimorphic seeds of *S. liaotungensis* in light conditions. There were 50 seeds counted for each replicate. Different upper-case letters show significant differences in germination percentage for the same seed type and temperature between light and darkness conditions (*t*-test, *p* < 0.05). Different lower-case letters show significant differences in germination percentage, germination index, or germination potential at different temperatures for the same seed type (LSD test, *p* < 0.05).

**Figure 6 plants-12-01408-f006:**
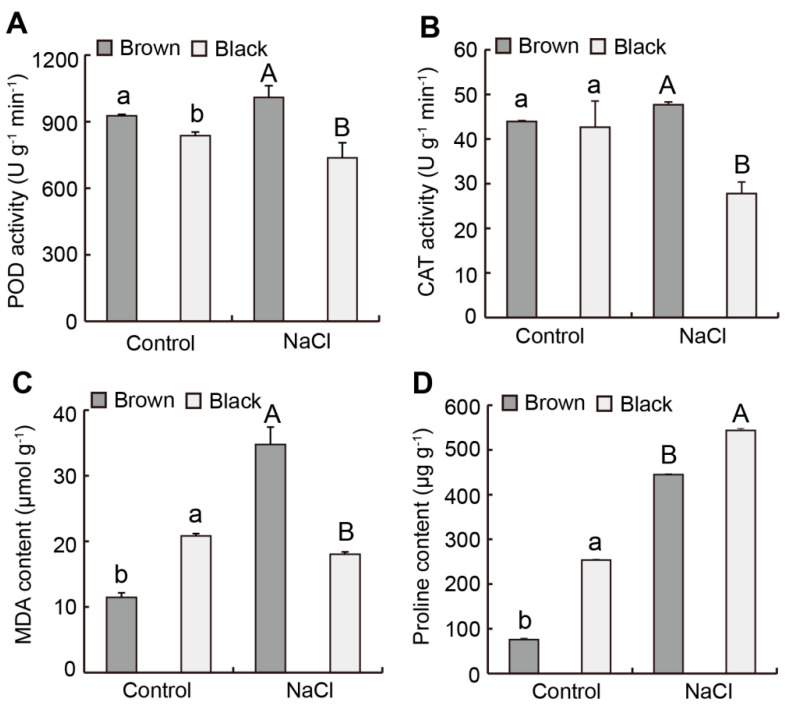
Effect of NaCl on the physiological indexes of dimorphic seeds during germination in *S. liaotungensis*. (**A**) POD activity. (**B**) CAT activity. (**C**) MDA content. (**D**) Proline content. Different lower-case letters show significant differences between dimorphic seeds during germination in the absence of NaCl solution (*t*−test, *p* < 0.05). Different upper-case letters show significant differences between dimorphic seeds during germination in the presence of 400 mM NaCl solution (*t*-test, *p* < 0.05).

**Figure 7 plants-12-01408-f007:**
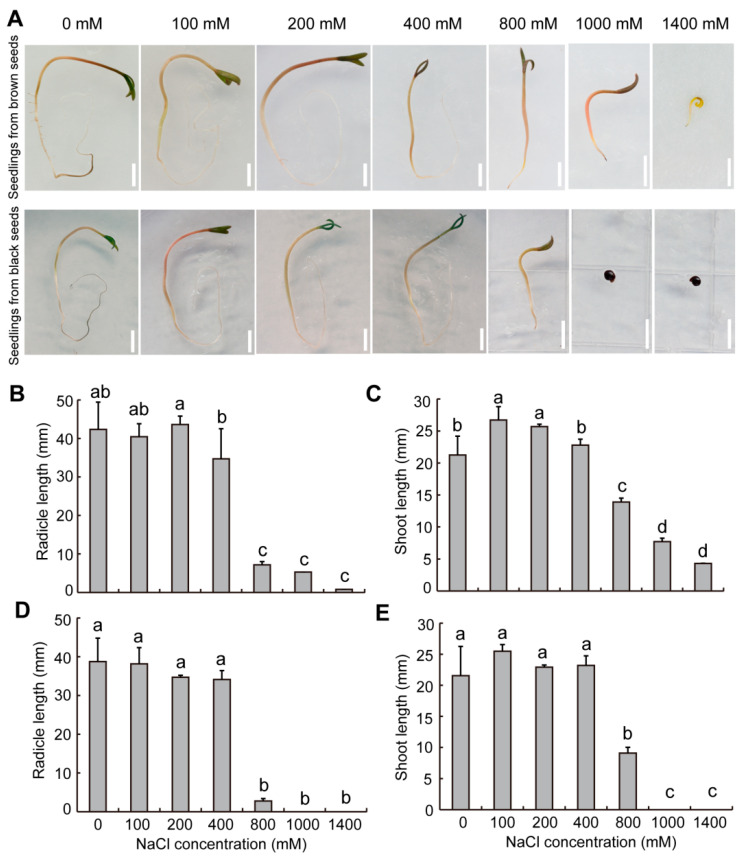
Effect of NaCl on the growth of seedlings from dimorphic seeds of *S. liaotungensis*. (**A**) Seedlings from dimorphic seeds under different NaCl concentrations’ treatment. Bars = 5 mm. (**B**,**C**) Radicle and shoot lengths of seedlings from brown seeds. (**D**,**E**) Radicle and shoot lengths of seedlings from black seeds. There were 10 seedlings counted for each replicate. Different letters show significant differences (LSD test, *p* < 0.05).

**Figure 8 plants-12-01408-f008:**
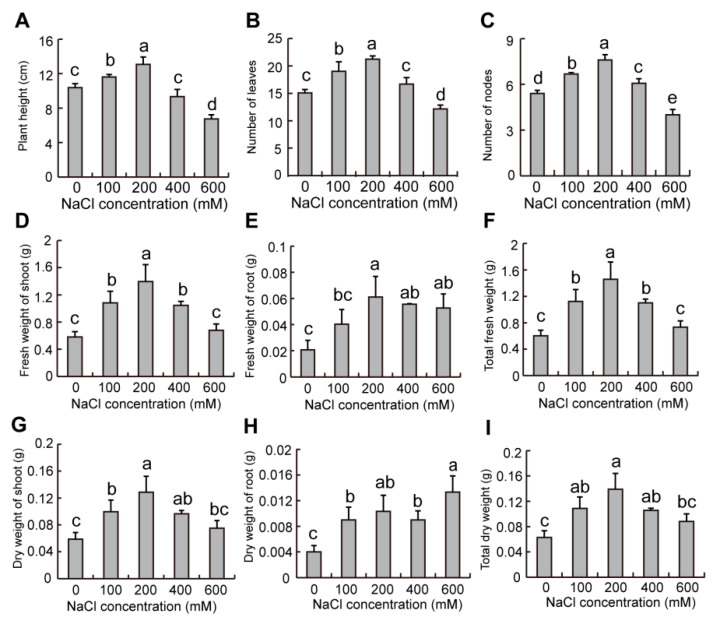
Effects of NaCl on the morphological indicators of plants from brown seeds of *S. liaotungensis*. The determination of plant height (**A**), numbers of leaves (**B**) and nodes (**C**), fresh weight of shoot (**D**), root (**E**), and total (**F**), and dry weight of shoot (**G**), root (**H**), and total (**I**) of plants from brown seeds. There were 5 plants counted for each replicate. Different letters show significant differences (LSD test, *p* < 0.05).

**Figure 9 plants-12-01408-f009:**
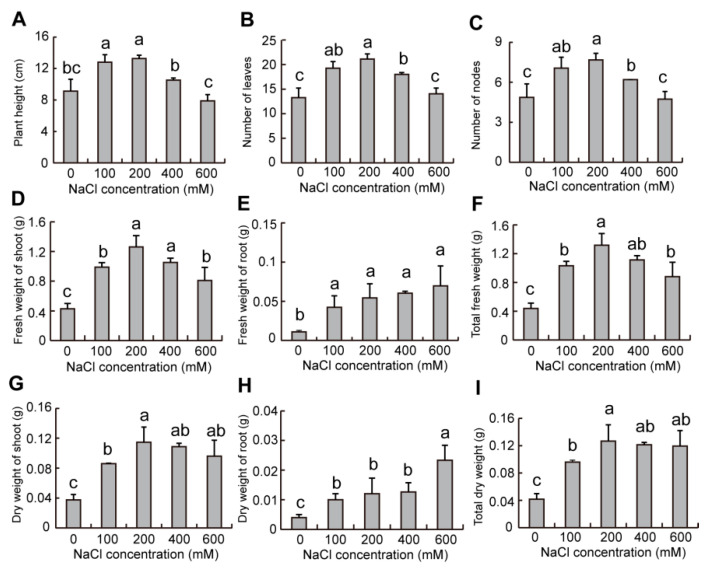
Effects of NaCl on the morphological indicators of plants from black seeds of *S. liaotungensis*. The determination of plant height (**A**), numbers of leaves (**B**) and nodes (**C**), fresh weight of shoot (**D**), root (**E**), and total (**F**), and dry weight of shoot (**G**), root (**H**), and total (**I**) of plant from black seeds. There were 5 plants counted for each replicate. Different letters show significant differences (LSD test, *p* < 0.05).

**Table 1 plants-12-01408-t001:** Effect of NaCl on germination of dimorphic seeds of *S. liaotungensis*.

NaCl Concentration (mM)	Brown Seed			Black Seed		
	Germination Percentage (%)	Germination Recovery Percentage (%)	Final Germination Percentage (%)	Germination Percentage (%)	Germination Recovery Percentage (%)	Final Germination Percentage (%)
0	100.0 ± 0.0 Aa	0.0 ± 0.0 Ac	100.0 ± 0.0 Aa	49.3 ± 4.6 Ba	0.0 ± 0.0 Ac	49.3 ± 4.6 Ba
100	96.0 ± 6.9 Aa	0.0 ± 0.0 Ac	96.0 ± 6.9 Aab	44.0 ± 2.0 Ba	0.0 ± 0.0 Ac	44.0 ± 2.0 Ba
200	98.0 ± 0.0 Aa	0.0 ± 0.0 Ac	98.0 ± 0.0 Aab	52.7 ± 7.0 Ba	0.7 ± 1.2 Ac	53.3 ± 6.1 Ba
400	89.3 ± 4.2 Aa	0.0 ± 0.0 Ac	89.3 ± 4.2 Ab	26.7 ± 18.1 Bb	22.7 ± 13.3 Bb	49.3 ± 5.0 Ba
800	61.3 ± 7.6 Ab	32.0 ± 10.6 Ab	93.3 ± 3.1 Ab	4.0 ± 5.3 Bc	45.3 ± 1.2 Aa	49.3 ± 4.2 Ba
1000	36.0 ± 9.2 Ac	48.7 ± 5.8 Aa	84.7 ± 4.2 Ab	1.3 ± 1.2 Bc	49.3 ± 10.1 Aa	50.7 ± 9.0 Ba
1400	19.3 ± 12.1 Ad	50.7 ± 9.0 Aa	70.0 ± 11.1 Ac	0.0 ± 0.0 Bc	43.3 ± 3.1 Aa	43.3 ± 3.1 Ba

The germination percentage, germination recovery percentage, and final germination percentage of dimorphic seeds of *S. liaotungensis* under different NaCl concentrations. There were 50 seeds counted for each replicate. Different lower-case letters show significant differences in germination percentage, germination recovery percentage, or final germination percentage for the same seed type at different NaCl concentrations (LSD test, *p* < 0.05). Different upper-case letters show significant differences between dimorphic seeds for the same concentration (*t*-test, *p* < 0.05).

**Table 2 plants-12-01408-t002:** Germination index of dimorphic seeds from *S. liaotungensis* under different NaCl concentrations.

NaCl Concentration (mM)	Brown Seed	Black Seed
0	46.2 ± 0.6 Aa	19.8 ± 2.3 Ba
100	43.3 ± 3.1 Aa	18.4 ± 0.7 Bab
200	35.2 ± 5.5 Ab	14.6 ± 3.1 Bb
400	26.2 ± 1.2 Ac	3.9 ± 4.1 Bc
800	12.6 ± 2.3 Ad	0.5 ± 0.6 Bcd
1000	7.4 ± 2.8 dAe	0.2 ± 0.2 Bcd
1400	5.3 ± 4.2 Ae	0.0 ± 0.0 Ad

There were 50 seeds counted for each replicate. Different lower-case letters show significant differences in germination index for the same seed type at different NaCl concentrations (LSD test, *p* < 0.05). Different upper-case letters show significant differences in germination index between dimorphic seeds for the same concentration (*t*-test, *p* < 0.05).

**Table 3 plants-12-01408-t003:** Radicle tolerance index of seedlings from dimorphic seeds under different NaCl concentrations.

NaCl Concentration (mM)	Seedlings from Brown Seeds	Seedlings from Black Seeds
0	100.0 ± 0.0 Aa	100.0 ± 0.0 Aa
100	96.5 ± 8.3 Aab	99.4 ± 12.4 Aa
200	104.4 ± 12.1 Aa	90.8 ± 11.8 Aa
400	81.8 ± 12.2 Ab	89.1 ± 10.8 Aa
800	15.9 ± 0.3 Ac	6.8 ± 0.4 Bb
1000	11.8 ± 1.5 Ac	0.0 ± 0.0 Bb
1400	1.8 ± 0.1 Ac	0.0 ± 0.0 Bb

There were 10 seedlings counted for each replicate. Different upper-case letters show significant differences between dimorphic seeds at the same NaCl concentration (*t*-test, *p* < 0.05). Different lower-case letters show significant differences in radicle tolerance indexes of seedlings for the same seed type at different NaCl concentrations (LSD test, *p* < 0.05).

**Table 4 plants-12-01408-t004:** Shoot tolerance index of seedlings from dimorphic seeds under different NaCl concentrations.

NaCl Concentration (mM)	Seedlings from Brown Seeds	Seedlings from Black Seeds
0	100.0 ± 0.0 Ab	100.0 ± 0.0 Aa
100	126.6 ± 11.8 Aa	121.1 ± 19.8 Aa
200	122.3 ± 14.4 Aa	109.3 ± 20.2 Aa
400	108.2 ± 10.7 Ab	109.9 ± 15.2 Aa
800	63.5 ± 7.5 Ac	39.5 ± 4.5 Bb
1000	35.1 ± 3.3 Ad	0.0 ± 0.0 Bc
1400	19.7 ± 2.7 Ad	0.0 ± 0.0 Bc

There were 10 seedlings counted for each replicate. Different upper-case letters show significant differences between dimorphic seeds at the same NaCl concentration (*t*-test, *p* < 0.05). Different lower-case letters show significant differences in shoot tolerance indexes of seedlings for the same seed type at different NaCl concentrations (LSD test, *p* < 0.05).

## Data Availability

Not applicable.

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
