# Peer review of "Differences in Physiological Characteristics, Seed Germination, and Seedling Establishment in Response to Salt Stress between Dimorphic Seeds in the Halophyte Suaeda liaotungensis"

_plants, 2023, doi:10.3390/plants12061408_

Round 1

Reviewer 1 Report

The manuscript presents interesting results related to the dimorphism of seeds of Suaeda liaotungensis, something that has been observed in other halophytes. Research brings new and relevant data. However, information is missing in the Material and Methods Section, which may be useful to improve the discussion. Specific comments were added to PDF file (Manuscript)

Reviewer 2 Report

The manuscript of Song et al. describes a research about differential physiological performances of dimorphic seeds of the halophyte Suaeda liaotungensis. Authors performed comparative studies on seed morphology, ROS levels, MDA production, antioxidant activities and osmolyte levels. Additionally, the effect of light, temperature and salinity on seed germination, and the effects of salinity in seedling and plant growth. The goal is to gain a better understanding of the adaptation strategies of dimorphic seeds to salinity and other environmental factors.

A similar work was previously performed on Suaeda salsa with salinity (Ref. 20), although no physiological-biochemical parameters were measured. In the present manuscript, the scientific objective is interesting, and methods and results are well done. However, in my opinion, this work needs to be improved in several aspects for publication.

Regarding research:

In order to accomplish the first question proposed by the Authors in the Introduction (what is the relationship between physiological indexes and salt tolerance in dimorphic seeds?), they quantified ROS, lipid peroxidation (MDA), osmolytes and antioxidant enzyme activities in both dimorphic resting seeds. In my opinion, further experiments must be performed. It would have been interesting to compare these physiological-biochemical values between control (absence of salinity) and salinity-exposed germinating dimorphic seeds (and even in seedlings). Such experiments would have provided more information for the question proposed by the Authors, which is the mean focus of the research.

Another option is to perform a comparative metabolomic or transcriptomic profiling, comparing dimorphic seeds (germinating in absence or presence of salt). Results of this experiment would elucidate the metabolic pathways involved in the differential responses of dimorphic seed germination to salt stress.

Regarding the writing of the manuscript, several points need to be corrected:

-    The updated family name is Amaranthaceae, not Chenopodiaceae. This must be corrected in several sentences in the manuscript.

-      Discussion about proline results must be included.

-  In Discussion, authors say that “MDA might act as a negative regulator in salt tolerance”. MDA is a product of lipid peroxidation and oxidative damage by ROS. You can say that it can be used as a marker or indicator of salt tolerance, but not it is not a “regulator” in salt tolerance. Please correct.

-  In Discussion, I do not understand what authors mean by “germination space”.  Please, explain or correct.

-     In several parts of the manuscript, Authors refer to “progeny plants” from dimorphic seeds. But here, “progeny” is a wrong Word. Authors worked with seeds, and with seedlings and plants germinated from those seeds, they did not work with the F1 (or progeny proper) of those seeds. Please change the Word “progeny” in those sentences, in order to avoid confusion.

-  In Material and Methods, section 4.1 (Plant materials), a lot of information is missing. What are the climatic and/or environmental characteristics of the sampled place? That is, what are the temperature, light and water regime along the year???, What is the Suaeda (and other plant species) population and distribution?? Suaeda life cycle along the year?, When this plant does germinate dimorphic seeds (what months along the year)?? What is the brown seeds/black seeds number rate in the sampled population?? It also lacks the soil physico-chemical characterization (organic matter, nutrients, salinity, pH, texture??) in the sampled place. Most of these data are very relevant and would provide important information to contextualize the results. If this information has been previously published elsewhere, please add references.

-    In Materials and Methods, section 4.4 (seed germination), Authors say that seeds were incubated “under light condition or constant darkness”. Please, explain what was the light condition (light type and intensity).

- English should be revised in the manuscript. Several sentences and expressions along the manuscript are confusing and not grammatically correct.

-  Additional comments and corrections are labelled and included in the PDF manuscript, that I send as attached file.

To conclude, in my opinion, this research requires additional experiments for publication in PLANTS, comparing germinating dimorphic seeds/seedlings in absence and presence of salinity, in order to elucidate the relationship between seed physiology and salt tolerance.

Reviewer 3 Report

General comments:

The present manuscript focuses on differential responses of seed germination and seedling growth to salt stress in the halophyte Suaeda liaotungensis. The results gave an in-depth understanding of the adaptation strategies of dimorphic seeds to salinization environment, and better exploitation and utilization of S. liaotungensis. The manuscript holds scientific potential but before final publication, some suggestions need to be addressed to improve the overall quality of the manuscript.

Moderate English editing is required.

Suggestions for authors:

Title: Modify the title a little bit, it is confusing.

Abstract: The abstract should be started with the current problem and why the authors have chosen this study. Rather than general statements, the authors should highlight the results.

Introduction: It is quite lengthy. Also, there should be a connection between different subsections to highlight the importance of the current study.

Rectify the spacing error throughout the manuscript, especially between the references.

Section 2.1. It should not be bold as per the journal’s format. It should be followed throughout the manuscript.

Discussion: It is very lengthy and disoriented. It should be precise with supporting literature.

Which standard protocol was used for seed germination protocol?

Materials and methods: The methods should be shortened. Rather they seem disoriented. Modify them.

Conclusions: It should be precise and must highlight the overall findings of the present manuscript.

Round 2

Reviewer 2 Report

I found that Authors have improved the clarity and quality of the manuscript after addition of most suggestions proposed by the reviewers. However, it still remains my major concern about this work, that is, additional results comparing physiological-biochemical or molecular parametres (ROS, MDA, osmolytes, antioxidant enzymatic activities, or alternatively, comparative transcriptomics) in seeds germinating in absence and presence of salinity are required to accomplish the goal of the proposed research. In their reply, Authors have not commented nor added additional results in this regard.

Round 3

Reviewer 2 Report

Results of these Authors about transcriptome analysis of brown seeds during germination, that they have already published, are interesting. But the present manuscript is about differences between dimorphic brown and black seeds. So it still lacks a comparative study of dimorphic seeds during germination in absence and presence of salinity. Instead of transcriptomics, comparative studies could be about biochemical parameters (proline levels, MDA, antioxidant enzymes). Of course, transcriptomic comparison would be much more informative.
